# Transcriptome Analysis of White- and Red-Fleshed Apple Fruits Uncovered Novel Genes Related to the Regulation of Anthocyanin Biosynthesis

**DOI:** 10.3390/ijms25031778

**Published:** 2024-02-01

**Authors:** Sylwia Keller-Przybylkowicz, Michal Oskiera, Xueqing Liu, Laiqing Song, Lingling Zhao, Xiaoyun Du, Dorota Kruczynska, Agnieszka Walencik, Norbert Kowara, Grzegorz Bartoszewski

**Affiliations:** 1The National Institute of Horticultural Research, Konstytucji 3-go Maja, 96-100 Skierniewice, Poland; michal.oskiera@inhort.pl (M.O.); agnieszka.walencik@inhort.pl (A.W.); norbert.kowara@inhort.pl (N.K.); 2Yantai Academy of Agricultural Science, Gangechengxida Street No 26, Fushan District, Yantai 265500, China; lxueqing@126.com (X.L.); zhaolingling122@163.com (L.Z.); 15726568706@126.com (X.D.); 3Department of Plant Genetics Breeding and Biotechnology, Warsaw University of Life Sciences, Nowoursynowska 159, 02-776 Warsaw, Poland; grzegorz_bartoszewski@sggw.edu.pl

**Keywords:** apple, expression profiling, KEGG, red-flesh fruits, RNA-seq, RT-qPCR, transcriptome

## Abstract

The red flesh coloration of apples is a result of a biochemical pathway involved in the biosynthesis of anthocyanins and anthocyanidins. Based on apple genome analysis, a high number of regulatory genes, mainly transcription factors such as MYB, which are components of regulatory complex MYB-bHLH-WD40, and several structural genes (*PAL*, *4CL*, *CHS*, *CHI*, *F3H*, *DFR*, *ANS*, *UFGT*) involved in anthocyanin biosynthesis, have been identified. In this study, we investigated novel genes related to the red-flesh apple phenotype. These genes could be deemed molecular markers for the early selection of new apple cultivars. Based on a comparative transcriptome analysis of apples with different fruit-flesh coloration, we successfully identified and characterized ten potential genes from the plant hormone transduction pathway of auxin (*GH3*); cytokinins (*B-ARR*); gibberellins (*DELLA*); abscisic acid (*SnRK2* and *ABF*); brassinosteroids (*BRI1*, *BZR1* and *TCH4*); jasmonic acid (*MYC2*); and salicylic acid (*NPR1*). An analysis of expression profiles was performed in immature and ripe fruits of red-fleshed cultivars. We have uncovered genes mediating the regulation of abscisic acid, salicylic acid, cytokinin, and jasmonic acid signaling and described their role in anthocyanin biosynthesis, accumulation, and degradation. The presented results underline the relationship between genes from the hormone signal transduction pathway and *UFGT* genes, which are directly responsible for anthocyanin color transformation as well as anthocyanin accumulation during apple-fruit ripening.

## 1. Introduction

Apples are one of the most widely cultivated crops in the world’s temperate regions. Their cultivation is very popular, especially due to their high economic value and the nutritional value of the fruits [1,2]. The annual global production of apples is about 78.4 million tons. Poland is the largest producer of apple fruits in the European Union (4.3 million tons), third in the world after China (35–40 million tons) and the USA (over 4.5 million tons) [3]. The management of such massive fruit production raises challenges in terms of cultivation and storage. However, an opportunity to address this situation lies in increasing the consumption of fresh apples and their products. This can be achieved by introducing into the apple industry red-fleshed varieties, which are rich in anthocyanins and have health-promoting properties [4]. In addition, apples with red flesh are an interesting raw material for the production of cloudy apple juices [5].

Intensive apple breeding efforts allowed us to confirm that *Malus sieviersii* is the most closely related to the commonly cultivated ‘Golden Delicious’ cultivar. Moreover, the primary donors of genes for the red skin and red flesh of apple fruits are *M. sieversii* f. *niedzwetzkyana* and *M. pumila* var. *niedzwetzkyana* [6]. As red fruits are highly attractive to consumers and often associated with apple ripeness and good flavor, their pigmentation becomes an important trait in horticultural research [2,7]. However, there are only a few breeding programs focused on developing new red-fleshed cultivars with high-quality fruits [8]. An analysis of apple segregating F_1_ populations combined with QTL mapping and transcriptomic studies has highlighted the complexity of this trait [9,10,11,12].

Anthocyanins (glycosides of anthocyanidins), which are synthesized by the flavonoid pathway, are the most important fruit component contributing to red-skinned and red-fleshed apple fruits and red apple plant tissues (e.g., stems, leaves, and seeds). They have been extensively studied, especially in regard to human health, playing a huge role in reducing the risks of cardiovascular disease, asthma, gastrointestinal diseases, etc., and they also contribute to weight regulation [11,13,14]. The flavonoid pathway is well-recognized and consists of a number of different enzymes that catalyze the biosynthesis of anthocyanidins and anthocyanins [15]. In apples, anthocyanidins are divided into several subclasses: pelargonidin, cyanidin, delphinidin, peonidin, petunidin, and malvidin, and most of them are cyanidins; thus, the reconstruction of galactose into the cyanidin form galactoside is mainly based on the transformation of galactose into cyanidin-3-galactoside [16]. The determination of anthocyanidin coloration in specific tissues is influenced by different types and numbers of substituents in the anthocyanin ring [8,17,18,19]. 

The initial precursor in the anthocyanins and flavonoids biosynthesis pathway is phenylalanine. It is subsequently catalyzed by phenylalanine ammonialyase (*PAL*), 4-coumarate:coenzyme A ligase (*4CL*), chalcone synthase (*CHS*), chalcone isomerase (*CHI*), flavanone 3-hydroxylase (*F3H*), flavonoid 3′5′-hydroxylase (*F3*′*5*′*H*), and dihydroflavonol 4-reductase (*DFR*), leading to the production of colorless metabolites. Then, the colored anthocyanins are synthesized by anthocyanidin synthase (*ANS*). Finally, they are transformed into brick-red, magenta, or even blue–violet glycosides through the addition of UDP-glucose by flavonoid 3-O-glucosyltransferase (*UFGT*) [20,21]. 

Based on apple genome analysis, several genes encoding MYB transcription factors involved in anthocyanin biosynthesis regulation were identified [8,22,23,24]. The overexpression of *MYB10* is strongly correlated with the regulation of type 1 apple flesh color through anthocyanin increments in skin, shoots and fruit flesh (expressing the pigmentation of fruit flesh, cortex and a white core). This can be a result of the autoregulation of *MYB10*, leading to the red-flesh apple phenotype [17,25,26]. Meanwhile, in type 2, the red-fleshed apple fruit phenotype did not co-segregate with *MYB10* [8]. An analysis performed by Chagné et al. [10] and Mahmoudi et al. [20] confirmed the relationship between *MYB* allele combinations and fruit-flesh color. The characterization of 16 apple cultivars with red and white flesh, as well as the segregating population derived from the cross of ‘Geneva’ and ‘Braeburn’, confirmed that the seedlings possessing an *MYB1*/*MYB1* allele combination had a red-skin and white-fruit-flesh phenotype, while the presence of *MYB10* alleles (*MYB1*/*MYB10* or *MYB10*/*MYB10*) shaped the genotypes with red skin, red fruit flesh, and red seeds [10,20].

Recent studies, established by the analysis of a different set of apple genotypes as well as the transgenic ‘Royal Gala’, allowed for investigating new genes such as *MdJa2*, *MdNAC*, *WD40*, *bHLH* and *SEPALLATA*, and *MADS-box* transcription factors, involved in brassinosteroids accumulation and affecting anthocyanin and proantocyanidins (PA) biosynthesis. MYB-bHLH-WD40 (MBW; a composition of R2R3-MYB and WD transcription factors) was described as a part of the transcriptional complex of the regulation of anthocyanin structural genes [7,12,17,27,28]. Simultaneously, current studies have also confirmed that transcription factors such as MYB17, MYB111, MYBL2 and MYB16, interacting with HLH33 and HB1 may negatively regulate anthocyanin biosynthesis [29,30]. 

Since the mechanism of regulation of red-fleshed apple fruits is complex and still unclear, the objective of this study was to investigate new, so far unrecognized, genes involved in apple fruit-flesh coloration. These genes could become candidates for developing reliable molecular markers to accelerate the early selection of apple cultivars with red flesh that are applicable in apple breeding programs.

## 2. Results

### 2.1. Transcriptome Profiling 

The major goal of this analysis was to determine the difference between gene transcription in fruits of red- and white-fleshed apple cultivars. An RNA-seq experiment with fruits of Red Love^®^ ‘General’ and ’Early Fuji’, characterized by red and white fruit flesh, respectively, was performed. For each cultivar, three independent fruit samples (three replications) were used; six RNA-seq libraries were constructed; and over 40 G of clean reads were obtained. The effective data volume for each sample was in the range of 6.06 to 7.23 G. The quality of reads was high, with a Q30 ratio from 92.33 to 92.78% of inferred base call accuracy. The GC content was similar in all samples, with an average of 47.46% (Appendix A). Clean reads were mapped on the apple reference genome at a similar ratio, 93.63% on average, with 89.7% being unique mapping reads and 85.91% being reads mapped in proper pairing. 

The expression of protein-coding genes was calculated using the FPKM method, which indicates the number of fragments per kilobase length of a protein-coding gene per million sequenced fragments. For 35,890 genes annotated in the apple genome, a similar number in each analyzed sample, on average 25,666 genes, was found to be expressed (Appendix A). A sample-to-sample cluster analysis and principal component analysis (PCA) of expression profiles indicated that the results were consistent, and investigated cultivars samples were clearly grouped together (Figure 1).

### 2.2. Differentially Expressed Genes

Based on gene expression profiling, differentially expressed genes (DEGs) in Red Love^®^ ‘General’, as compared to ‘Early Fuji’ (control), were identified (at a log2 fold-change > 1 and a false discovery rate (FDR) of <0.05) (Appendix A). In total, 4839 genes were found to be differentially expressed in Red Love^®^ ‘General’, with 2259 genes up-regulated and 2580 down-regulated (Figure 2). 

Based on the filtered data, a Venn diagram was prepared to indicate the number of up-regulated genes with a log2 fold-change > 2 (2259 DEGs) and a log2 fold-change > 3 (1425 DEGs), as well as down-regulated genes with a log2 fold-change > 2 (2580 DEG’s) and a log2 fold-change > 3 (1706 DEGs). According to the log2 fold-change > 2, 74 down-regulated and 834 up-regulated unique genes were revealed (Figure 3).

### 2.3. Functional Categories of DEGs 

#### 2.3.1. GO Enrichment Analysis of DEGs

After finding the differentially expressed genes in red-fleshed Red Love^®^ ‘General’, we performed Gene Ontology (GO) enrichment analysis to describe their functions.

The GO enrichment analysis allowed for classifying the general role of differentially expressed genes in biological processes (BP), cellular component integration (CC) and molecular function (MF). The total GO enrichment categorization of 3500 DEGs (Appendix A) led to deeming 1635 genes as up-regulated and 1865 as down-regulated in red-fleshed fruits of Red Love^®^ ‘General’ as compared to white-fleshed ’Early Fuji’ (Appendix A). Most of the DEGs were clustered into 65 different GO terms: 23 for biological process, 20 for cellular component integration and 21 for molecular function. In terms of biological process, the highest number of DEGs was assigned to the categories: biological regulation, cellular process, metabolic process, response to stimulus and single-organism process. In terms of cellular component integration, the highest number of DEGs was assigned to the categories: cell and cell part, membrane and membrane part integration, and organelle and organelle part. In terms of molecular function, most genes were assigned to two categories: binding and catalytic activity.

Next, DEGs from the top 30 GO enriched categories were analyzed (Appendix A). In the main category of biological process, the top up-regulated DEGs (Appendix A) were classified in terms of: GO:0042445—hormone metabolism process, GO:0008202—steroid metabolic process, GO:0009850—auxin metabolic process and GO:0009684—indoleacetic acid biosynthesis. There were other terms additionally enriched: GO:0023052—signaling, GO:0050896—response to stimulus, GO:0044700—single organism signaling, GO:0006952—defense response, GO:0007165—signal transduction, GO:0007154—cell communication, GO:0034050—host programmed cell death, and GO:0009626—plant hypo-sensitivity. In the category of cellular component, two top GO enrichment terms were extracted: GO:0071944—cell periphery and GO:0005886—plasma membrane (cell membrane integration) (Appendix A). In the category of molecular function, the top GO enrichment terms were assigned to: GO:0043531—ADP binding and GO:0016491—oxidoreductase activity (Appendix A).

#### 2.3.2. KEGG Enrichment Analysis of DEGs

An analysis of significantly enriched pathways according to the Kyoto Encyclopedia of Genes and Genomes (KEGG) revealed that the highest numbers of DEGs were classified into six main groups: cellular processes, environmental information processing, genetic information processing, human diseases, metabolism, and organismal systems (Figure 4a). The biggest group included genes involved in metabolic pathway regulation (in total 812 DEGs; blue blocks), and the second group was involved in genetic information processing (in total 205 DEGs; green blocks). In addition, the third group of genes was classified into pathways of plant metabolism in diverse environments (in total 99 DEGs; brown blocks). The subgroups of DEGs classified based on metabolism were assigned to: carbohydrate metabolism (170), amino acid metabolism (110), energy metabolism (105), and lipid metabolism (99). For the other groups, the highest number of DEGs (90) was classified according to signal transduction (Figure 4b). The extracted up- and down-regulated genes recovered from Red Love^®^ ‘General’ vs. ‘Early Fuji’ compartments are presented in Appendix A. From this grouping, thirty top GO terms were uncovered: three of the most abundant from the biological process group were assigned to the galacturonate biosynthetic process and signal transduction; from the cellular components group, they were assigned to the plasma membrane and extracellular space, and from the group of molecular functions, they were assigned to genes involved in ADP binding and representing chalcone isomerase activity (Figure 4b).

A detailed analysis of the KEGG pathways enrichment classification showed that the group of DEGs most significantly up-regulated in red-fleshed Red Love^®^ ‘General’ was assigned to the plant hormone signal transduction pathway (mdm04075) (Figure 5). Then, the next ten up-regulated DEGs from this group, involved in auxin, cytokinine, gibberellin, abscisic acid, brassinosteroids, jasmonic acid, and salicylic acid hormonal signaling and regulation, were selected for further study (Figure 6). 

The functions and cell localization of selected genes are presented in Table 1 and the selected genes were addressed for detailed expression profiling in the apple cultivars characterized by red-fleshed fruits. 

### 2.4. Validation of Activity of Genes from Plant Hormone and Signal Transduction Pathway by RT-qPCR

The expression profiles of selected genes were analyzed in immature and ripe fruits of seven red-fleshed apple cultivars in comparison to the immature and ripe, white-fleshed fruits of ‘Free Redstar’ (Figure 7). 

Overall, when compared to ‘Free Redstar’, a significant repression of four out of ten selected genes—DELLA (involved in gibberellin pathway), BZR1 and TCH4 (brassinosteroids pathway), and MYC2 (responsible fo jasmonic acid transduction)—was observed in immature fruits of: ‘Trinity’, ‘Alex Red’, Red Love^®^ ‘Era’, Red Love^®^ ‘Circe’, ‘Roxana’ and ‘Red Love^®^ ‘Sirena’ (Figure 8). Interestingly, the most significant up-regulation of those genes was evaluated for *M. sieviersii* f. *niedzwetzkyana* (characterized for its red coloration of leaves, flower petals, core, stem, seeds, skin and flesh), representing a wild apple variety, and considered as the donor of target genes. Simultaneously, a relevant overexpression of the same set of genes was noted for control immature white-fruit-flesh samples of ‘Free Redstar’. In the case of ABF (involved in the abscisic acid pathway), a significant up-regulation was observed for all red-fleshed apple cultivars tested.

In the case of ripe fruits of red-fleshed cultivars, all genes applied in this study showed significant variation in their expression profiles. The activity of GH3, SnRK2, BRI1 and TCH4 genes was significantly higher in fruit samples of ‘Alex Red’, Red Love^®^ ‘Circe’ and Red Love^®^ ‘Sirena’. In the case of ‘Trinity’, Red Love^®^ ‘Era’ and ‘Roxana’, four genes—B-ARR (cytokinine pathway), DELLA (gibberellin pathway), BZR1 (brassinosteroids pathway), and MYC2 (jasmonic acid pathway)—showed significant down-regulation. A high expression level of the ABF gene (abscisic acid pathway) (ranged between 10× and 50× the number of transcript fold-changes), in comparison to the ‘Free Redsar’ ripe, white-fruit-flesh control, was noted for all red-fleshed apple cultivars evaluated. Overall, the activity of analyzed genes was overexpression in immature and ripe fruits of *M. sieviersii* f. *niedzwetzkyana*; meanwhile, down-regulation of all investigated genes was observed in ripe fruits of white-fleshed ‘Free Redstar’ (Figure 9).

### 2.5. The Activity of the Structural Genes in Red-Fleshed Apple Cultivars

Six genes from the polyphenol and anthocyanin biosynthesis pathway were analyzed to confirm their expression in red-fleshed apple cultivars. A significantly high level of expression of all structural genes was noted in immature fruits of *M. sieviersii* f. *niedzwetzkyana* (i.e., showing about an 18-fold change for *F3H*) and ‘Free Redstar’ white-fleshed apple fruits (Figure 10). In the case of *DFR* and *UFGT*, negligible activity was noted for analyzed immature fruit-flesh samples. 

Additionally, *CHS*, *F3H* and *UFGT* genes were significantly up-regulated in ripe fruits of ‘Trinity’, Red Love^®^ ‘Circe’ and Red Love^®^ ‘Sirena’. In general, a significant up-regulation (fold-change ranged from 10 to 60×) of *PAL*, *CHS*, *ANS* and *F3H* (encoding enzymes for anthocyanin precursors synthesis) was observed for all ripe fruits of red-fleshed apples. As we could expect, the activity of tested genes was significantly lower in ripe fruits of white-fleshed ‘Free Redstar’ (Figure 11). Simultaneously, the light-red fruit-flesh color of ‘Roxana’ cv. may be the result of a significantly higher expression of the *PAL* (60-fold-change) precursor gene and relative inhibition of other structural genes, probably causing the breakdown of intensity of anthocyanin biosynthesis in the genome of this cultivar. Interestingly, a mechanism probably similar to this regulation was noted for white-fleshed ‘Free Redstar’ (Figure 11). Generally, in red-fleshed apple cultivars, four structural genes (*PAL*, *CHS*, *F3H*, *ANS*) were significantly up-regulated, while a high activity of *DFR* and *UFGT* genes was noted only for ‘Alex Red’, Red love^®^ ‘Circe’ and Red Love^®^ ‘Sirena’.

Additionally, the lower activity of all tested genes was observed for ripe, in comparison to immature, fruits of *M. sieversii* f. *niedzwetzkyana*.

### 2.6. Investigated Genes Showed Inconsistent Expression Profiles in Comparison to RNA-seq, but Significant Intergenic Correlation

Genes from the plant signal transduction pathway selected on the basis of the RNA-seq experiment were expressed at a high level in the fruits of Red Love^®^ ‘General’ cultivated in China. They were chosen for further expression profiling with regard to transcriptome analysis. However, in the validation studies performed on the red-fleshed apple collection, grown in Poland, variable and inconsistent expression profiles of those genes were observed.

Gene-to-gene correlation coefficients between structural and uncovered genes were evaluated individually for a set of immature and ripe fruits of analyzed apple cultivars. The assessment of interaction between structural genes and those selected from the hormone signal transduction pathway in immature fruits of all tested cultivars showed a significant correlation between, e.g., *DFR*, *F3H*, and *UFGT* and *B-ARR*, *NPR1*, *GH3, SnRK2*, and *BRI1* (Table 2a). In ripe fruits, the correlation was noted between all identified genes and the structural *UFGT* gene (encoding flavonoid 3-O-glucosyl transferase, the gene responsible for final glycosides transformation and their pigmentation). This may be explained by the fact that most of the structural genes from the anthocyanin pathway and the plant hormone signal transduction pathway are potentially activated at an early stage of fruit development and show significant intergenic correlation.

Generally, in ripe fruits, a significant correlation was observed between the expression of the chalcone synthase gene (*CHS*) and *B-AAR*, *GH3*, *SnRK2* and *TCH4* (involved in the pathways of cytokinins, auxin, salicylic acid and brassinosteroids, respectively); between the expression of gene-encoding phenylalanine ammonialyase (*PAL*) and the same set of genes; and between *UFGT* and all genes selected from the hormone transduction pathway (Table 2b). The expression of *B-ARR*, *GH3*, *SnRK2* and *TCH4* genes (from the cytokinine, auxin, abscisic acid and brassinosteroids pathways, respectively) showed a correlation with *CHS*, *PAL* and *UFGT* genes and could be considered as possible indicators of the regulation of glucoside color transformation in red-fleshed apple fruits ready for harvesting.

## 3. Discussion

In the presented study, utilizing transcriptome comparisons and gene expression analysis of two apple cultivars producing red- and white-fleshed fruits, we have revealed new genes putatively related to the regulation of anthocyanin biosynthesis. Among over 40 Gigabases of filtered sequence data, two major groups of genes mapped on the reference apple genome were recovered. One group of sequence reads (43%) was assigned to be involved in metabolic pathways, and a second (31%) was assigned to be involved in secondary metabolite biosynthesis. Based on differentially expressed genes (DEGs) from the top 20 KEGG Gene Ontology (GO) term classifications (Figure 5), we selected a set of 10 genes from the plant hormone signal transduction pathway involved in auxin (AUX), cytokinine (CK), gibberellin (GA), abscisic acid (ABA), brassinosteroids (BR), jasmonic acid (JA) and salicylic acid (SA) biosynthesis. Following an RT-qPCR analysis of the genes of interest, we observed their variable activity, with up-regulation noted for *GH3*, *SnRK2*, *ABF* and *TCH4* and down-regulation noted for *B-ARR*, *DELLA*, *BZR1*, *BRI1*, *MYC2* and *NPR1*. However, although all the selected genes showed up-regulation in the red-fleshed Red Love^®^ ‘General’ cultivar used in the RNA-seq experiment, the results obtained for the RT-qPCR, performed for seven cultivars studied, were not consistently uniform. A similar observation of inconsistencies between RNA-seq and RT-qPCR comprehensive analysis was reported by Everaert et al. [31]. The authors concluded that, depending on the analysis workflow, 15–20% of genes are usually considered as ‘non-concordant’ in regard to the results obtained with RNA-seq and RT-qPCR. Those ‘non-concordant’ genes are defined when both approaches yield differential expressions in opposing directions or when one method shows differential expression while the other does not [31].

An analysis was carried out for a representative set of seven red-fleshed apple cultivars collected at two fruit developmental stages and compared to white-fleshed cultivars. ‘Free Redstar’ allowed us to show that changes in gene expression are closely related to the fruit-ripening process [26,32,33]. All structural genes and the newly revealed *GH3* (auxin response), *SnRK2* and *ABF* (both involved in ABA signaling), as well as *TCH4* (BR pathway), were activated in the ripe, red-fleshed fruits. In accordance with fold-change calculations, *DELLA* (GA response), *B-ARR* (CK response), *BZR1* and *BRI1* (BR receptors), as well as *MYC2* and *NPR1* (genes from JA and SA signaling pathways, respectively), were defined as being significantly inhibited in the ripe, red-fleshed fruits. The putative mechanism of gene regulation of the plant hormone and transduction pathway is presented in the scheme below (Figure 12).

In accordance with the presented scheme, a similar observation was described by Su, Shi and coworkers [28,34]. They postulated that the structural genes from the anthocyanin biosynthesis pathway were up-regulated in apple calli, suppressing the expression of the brassinosteroids resistant gene (*BZR1*) (BR accumulation = anthocyanin pathway activation). They also reported that the interaction of *BZR1* with *MdJa2* modulates the target gene transcription and generally regulates plant response to brassinosteroids [28,34]. Based on an observation of apple calli development, Zheng and coworkers also underlined anthocyanin biosynthesis regulation by plant hormones, including brassinosteroids, which strictly influenced their accumulation [35]. Simultaneously, as reported by Su et al., the application of external BR also promoted fruit maturation and pro-anthocyanin synthesis in apple tissue [28,35]. Our observations led to the comparable conclusion that the inactivation of *BZR1* is positively correlated with the promotion of anthocyanin accumulation in red-fleshed, ripe apple fruits.

Brassinosteroids signaling is perceived by receptors *BRI1* and *BZR1* in the cell membrane. As recent research explains, it is one of the major factors (similar to bHLH) playing a key role in the regulation of BR gene expression [35]. Both BR receptor genes analyzed in the presented study showed significantly low expression in the series of red-fleshed fruit cultivars, probably leading to brassinosteroids accumulation, thus promoting anthocyanin synthesis. This selecting mechanism described in previous studies of tomato, cucumber, strawberry and grape indicates the key role of brassinosteroids in fleshy fruit development and ripening [33,36,37,38,39].

In parallel, based on our observations, we found a significant inactivation of the genes *NPR1* and *MYC2*, responsible for salicylic acid (SA) and jasmonic acid (JA) signal transduction, respectively, influencing the negative correlation with anthocyanin regulation in apple fruits. Therefore, our observation provides new insight into anthocyanin biosynthesis dependence on salicylic and jasmonic acid signal inhibition, which has not been reported so far. 

In this regard, previous research by Sohn et al. and summarized by Wang and Chang explained the role of JA in the regulation of secondary metabolites, plant defense response, and organ development [7,40]. Moreover, later studies by Qi, An, and coworkers showed that the activation of *MYB* and *bHLH* significantly correlated with anthocyanin accumulation in *A. thaliana* and it is promoted by complex mechanisms, including the degradation of a repressor protein (the so-called JA ZIM DOMAIN) [34,41,42]. Recent reports of Wang et al. also underline the role of jasmonic acid as a signaling factor influencing the *MYB24L* gene and affecting anthocyanin biosynthesis [24].

In the conducted research, the newly studied DEGs such as *DELLA* (gibberellin pathway, responsible for ubiquitin-mediated proteolysis) and *B-ARR* (cytokinin pathway), showed significant down-regulation in the red-fleshed, ripe fruits and up-regulation in the white fruit flesh of ‘Free Redstar’. The observations are similar to the results obtained by Nawaz and coworkers, which confirmed the up-regulation of histidine kinase and *DELLA* genes in ripe fruits of white-fleshed apple ‘Hanfu’ and its mutant. The authors explained that the level of the gene transcript number is genotype-dependent, resulting in apple fruit-flesh pigmentation intensity—from flush flesh and pinky flesh to dark-red flesh [33,43]. In *Arabidopsis* and grape, it was uncovered that the DELLA gene also mediates the environmental stimulation of anthocyanin biosynthesis [34].

Another finding of our work underlines a negative correlation between gene activity and auxin and cytokinin production (gene activation = fewer auxins; gene inactivation = more cytokinin production), expressed by the up-regulation of the auxin response gene (*GH3*) and down-regulation of *B-AAR* in red-fleshed cultivars. This gene activity and metabolite production relationship was highlighted by Ji and coworkers, who observed that increments of auxins together with a decrease in cytokinin in the cells can inhibit anthocyanin synthesis [44]. This phenomenon was observed in callus tissue of both *A. thaliana* and apple and underlined by Nguyen, Ji and coworkers, who elucidated that the increased concentration of cytokinin and decrease in auxin can significantly promote the expression of *MYB* transcription factors, leading to anthocyanin accumulation [45,46]. In addition, as reported by Shi et al., cytokinin enhances sucrose-mediated anthocyanin pigmentation and, especially under plant stress response, seems to play a negative role in anthocyanin accumulation [34].

Additionally, we have noted the up-regulation of *ABF* and *SnRK2* genes (involved in the abscisic acid pathway) in red-fleshed, ripe fruits. Notably, this was observed in all cultivars except ‘Roxana’ (pink fruit flesh) and *M. sieversii* f. *niedzwetzkyana* (a wild apple variety). This observation suggests a distinct regulation of the fruit-flesh coloration in these two cultivars. The gene activity in this context is genotype-dependent, highlighting the complexity of the regulation of apple fruit-flesh coloration. As described by Jiang and Joyce, ABA plays a key role in strawberry fruit color development (generally through PAL activity stimulation), and in grapes and litchi, it promotes the fruit coloration stage [47,48,49].

Through the effect of regulating the number of gene transcripts, we noted the changes in targeted hormonal pathways, which, by regulating the content of hormonal proteins, influence the mechanism of anthocyanin synthesis and accumulation in apple fruits. For the *DELLA, B-ARR*, *BZR1*, *BRI1*, *MYC* and *NPR1* genes, we observed a negative regulatory mechanism by gene repression, which thus resulted in hampering the recognition of or degrading hormonal proteins.

The uncovered genes were correlated with the functional gene *UFGT* (responsible for final anthocyanin color transformation). To date, only limited studies have examined the negative regulatory mechanisms underlying anthocyanin synthesis in apples. The novel technologies applied in our research allowed for understanding the mechanism of apple fruit color development. Uncovered genes, as candidates for molecular markers, can accelerate the breeding of high-quality, red-fleshed apple cultivars. The application of such research provides knowledge of the regulation of fruit color development and anthocyanin biosynthesis, accumulation, and degradation.

## 4. Materials and Methods

### 4.1. Plant Material

In the RNA-seq experiment, red-fleshed Red Love^®^ ‘General’ and white-fleshed ‘Early Fuji’ apple cultivars were used. Apple fruits were collected from trees grown in an experimental orchard of the Yantai Academy of Apple Research in China. Fruits (a minimum of three) of both cultivars were collected at harvest time (middle of September). From each fruit, peeled flesh discs with a diameter of 2 cm and a depth of 1 cm, cut with a sterile blade, were collected. The samples were placed directly into liquid nitrogen and stored at −80 °C until RNA extraction.

During the validation of gene expression experiments, a set of apple cultivars characterized by different patterns of fruit pigmentation was used. This set included: ‘Alex Red’, ‘Roxana’, ‘Trinity’, Red Love^®^ ‘Circe’, Red Love^®^ ‘Era’, Red Love^®^ ‘Sirena’ and *M. sieviersii* f. *niedzwetzkyana* and a control with white fruit flesh, ‘Free Redstar’ (flesh pigmentation control). Tissue samples (apple flesh from three fruits—biological replications) were collected from apple trees grown in the experimental orchard maintained at the National Institute of Horticultural Research in Skierniewice, Poland. For each cultivar, three fruits were collected at two developmental stages: immature fruits sampled 100 days after full bloom (DAFB) and ripe fruits collected at harvest time. Then, the apple fruit-flesh discs (as described above) were prepared and immediately placed in liquid nitrogen for RNA isolation. The samples were used for a reverse transcriptase quantitative PCR (RT-qPCR) gene expression analysis of target structural genes, as well as for differentially expressed genes revealed by RNA-seq.

### 4.2. RNA Extraction

For next-generation sequencing (NGS), total RNA from three independent fruits of Red Love^®^ ‘General’ and ‘Early Fuji’ was isolated using a MINIBEST Plant RNA Extraction Kit (TaKaRa, Dalian, China) according to the manufacturer’s protocol. The concentration and quality of RNA were evaluated after electrophoresis in 1% agarose gel stained with GelRed Dye (Biotium, Wuhan, China). High-quality RNA preparations were sent to a commercial company for transcriptome sequencing. 

For RT-qPCR, total RNA was isolated using the method described by Zeng and Yang [50] with minor modifications. The procedure was started with sample incubation for 20 min in CTAB buffer heated to 65 °C. After centrifugation in a mixture of chloroform and isoamyl alcohol (24:1, *v*/*v*), RNA (from three fruit replicates) was resolved from the upper solution fraction and precipitated with 10 M lithium chloride overnight at 4 °C. Then, the precipitate was dissolved in RNase-free water. The samples were treated with DNase I using a Turbo DNA-free kit (Thermo Fisher Scientific, Waltham, MA, USA) according to the manufacturer’s protocol. The quality, degree of integration, and concentration of RNA samples were assessed using the Agilent 2100 Bioanalyzer and Expert 2100 software: http://www.chem.agilent.com/ (accessed on 26 January 2024)) (Agilent Technologies, Santa Clara, CA, USA).

### 4.3. RNA-Seq Analysis

The library construction and sequencing were completed by Shanghai Ouyi Biomedical Technology Co., Ltd. (Shanghai, China). The transcriptome sequencing using Illumina technology was conducted by OE Biotech Co., Ltd. (Shanghai, China). To perform a quality check of the raw data and summarize the number of reads, Trimmomatic software http://www.usadellab.org/cms/index.php?page=trimmomatic (accessed on 26 January 2024) [51] was used. To map sequence reads on the apple reference genome, Htseq-count software http://www-huber.embl.de/HTSeq (accessed on 26 January 2024) [52] was used. As a reference, the ‘Golden Delicious’ apple genome assembly ASM211411v1 was applied: https://www.ncbi.nlm.nih.gov/data-hub/genome/GCF_002114115.1/ (accessed on 26 January 2024). The DESeq [53] allowed us to standardize the number of counts of each single gene, calculate the multiple of difference, and perform significant difference tests. To obtain the number of reads compared to protein-coding genes in each sample, Cufflinks [54] was applied. Finally, the gene expression level was calculated as FPKM (Fragments Per kb Per Million Reads), providing the length, as the number of fragments per kilobase, of a protein-coding gene per million mapped reads. The raw sequence data were stored in FASTQ file format. The evaluation of differentially expressed genes (DEGs) of Red Love^®^ ‘General’ and ‘Early Fuji’ was performed with use of DEseq2 R package software https://www.networkanalyst.ca/ (accessed on 26 January 2024). For this purpose, RNA-seq data were compared and analyzed, considering whether the same gene was differentially expressed within the two samples. For this purpose, log2FC > 1 (logarithm of fold-change in the expression level of the same gene in two analyzed samples), log2FC > 2 or log2FC > 3 and filtering data condition *p*-value < 0.05 criteria were applied. The RNA-seq datasets associated with this study were submitted to the National Center for Biotechnology Information (NCBI, Bethesda, MD, USA) Sequence Read Archive (SRA) database under the accession: SUB14156737 with the Bioproject ID: PRJNA1067510.

### 4.4. GO Enrichment Analysis

Gene function enrichment was performed based on the compilation of the list of all protein-coding genes and differential protein-coding genes (counted based on FPKM), using a hypergeometric distribution test to calculate the representative GO function set. Then *p*-value of significant enrichment in the list of differential protein-coding genes was corrected by Benjamini and Hochberg’s multiple tests to obtain a false discovery rate (FDR). This procedure translates experimental results into a series of regulated gene lists at multiple false discovery rate (FDR) cutoffs and computes the *p*-value of the overrepresentation of the genes.

The analysis of the top 30 GO enrichments was performed based on screened GO entries corresponding to the number of differential genes greater than two in the three categories (biological process (BP), cellular component integration (CC), and molecular function (MF)), which were sorted into ten entries according to the −log10 (*p* value) corresponding to each entry from a large to a small layout. 

The Fisher algorithm was used to conduct CC, BP, and MF enrichment analysis on the differential genes between samples [55], and topGO was used to draw a directed acyclic graph for the enriched term to visually display the GO term of differentially expressed gene enrichment and hierarchical (more specific) relationships. 

### 4.5. KEGG Enrichment Analysis of Differential Genes

The KEGG (Kyoto Encyclopedia of Genes and Genomes) database was screened for the pathway analysis of differential protein-coding genes. The hypergeometric distribution test was used to calculate the significance *p*-value of differential gene enrichment in each pathway entry. KEGG enrichment analysis of the top 20 pathway entries corresponding to the number of differential genes greater than 2, were sorted by the −log10 (*p* value) and presented by a bubble chart (Figure 5).

### 4.6. cDNA Synthesis and RT-qPCR

Total RNA (1 µg) was isolated from the tissue of three independent fruits collected from each apple cultivar and reverse-transcribed into cDNA using the AffinityScript QPCR cDNASynthesis Kit (Agilent, CA, USA). The reverse transcription reaction was carried out with a universal oligo-dT primer and reverse transcriptase (RT) in optimized thermal conditions: 25 °C for 5 min, 42 °C for 5 min (oligo-dT annealing), 55 °C for 15 min (reverse transcription), and 95 °C for 5 min (enzyme inactivation) in a Biometra Basic thermocycler (Biometra, Germany). For the differentially expressed genes uncovered in this study, oligonucleotides were designed using Primer3plus software (https://www.bioinformatics.nl/cgi-bin/primer3plus/primer3plus.cgi (accessed on 26 January 2024)). For structural genes encoding key enzymes of the anthocyanin biosynthesis pathway, oligonucleotides published by Kondo et al. [56] were used. Gene-encoding *ACTIN* [30] was used as an RT-qPCR data normalizer (Table 3).

RT-qPCR was performed with a Kapa SYBR qPCR kit in the presence of SYBR Green fluorescent dye (KapaBiosystems, Wilmington, MA, USA) using a RotorGen 6000 thermal cycler (Corbett Research, Mortlake, Australia). In each experimental setup, two pairs of specific primers, complementary to the evaluated structural gene and the selected target DEGs, were used in analogous reactions. The cDNA template was prepared in dilutions of known concentrations, enabling the preparation of a standard amplification reaction curve. The thermal profile of the RT-qPCR reaction was as follows: 95 °C for 5 min (polymerase activation), then 40 cycles including the steps: 95 °C for 15 s (denaturation), 60 °C for 20 s (oligonucleotide annealing), and 72 °C for 20 s (fluorescence level detection). Relative expression (fold-change) normalized in regard to *ACTIN* was determined on the basis of single data points derived from real-time PCR amplification curve threshold cycles (Ct values) (2^−∆∆Ct^) described by Livak and Schmittgen [57]. For this purpose, Rotor-Gene 6000 Series Software 1.7 was used (Corbett Research, Australia). The average of relative expression was normalized to the white-flesh fruit control ‘Free Redstar’. The standard error of the mean ± SEM and t-significance at the levels of *p*-values > 0.05 (*), 0.01 (**), 0.001 (***) between ‘Free Redstar’ and red-fleshed cultivars were calculated separately (GraphPad Prism 10.0.3, Dotmatics, Woburn, MA, USA). Relative fold-change diagrams for each gene were drawn using GraphPad Prism 10.0.3. The gene-to-gene correlation matrix between structural and newly uncovered genes was calculated using the Pearson correlation coefficient (r = 1, −1) with a *p*-value < 0.05 (GraphPad Prism 10.0.3 software).

## 5. Conclusions

In this research, we have carried out an evaluation of the expression profiles of genes related to the plant hormonal pathway in apple fruits. We defined a set of genes showing diverse activity in immature and ripe fruits of red-fleshed apple cultivars. The selected genes—*ABF*, *SnRK2*, *MYC2*, *NPR1* and *DELLA*—appear to be involved in the positive or negative regulation of plant signaling pathways. They may play a crucial role in mediating the abscisic acid, jasmonic acid, salicylic acid and gibberellin pathways, respectively, thereby influencing anthocyanin transformation, accumulation, and biosynthesis. All investigated genes showed a significant correlation with the *UFGT* gene in ripe, red-fleshed fruits. While there are many reports on the involvement of auxins, cytokinines and brassinosteroids in apple-fruit ripening and fruit-flesh pigmentation, the molecular basis for the regulation of other hormonal transduction factors and their impact on the constitution of red fruit flesh in apples have not been deeply investigated. Despite the preliminary research conducted, further analyses, including an assessment of important fruit quality parameters (such as fruit vitamin C, polyphenol, and anthocyanin values), are necessary to determine the identified genes for the early selection of red-fleshed apple varieties using molecular marker-assisted selection (MAS) procedures.

## Figures and Tables

**Figure 1 ijms-25-01778-f001:**
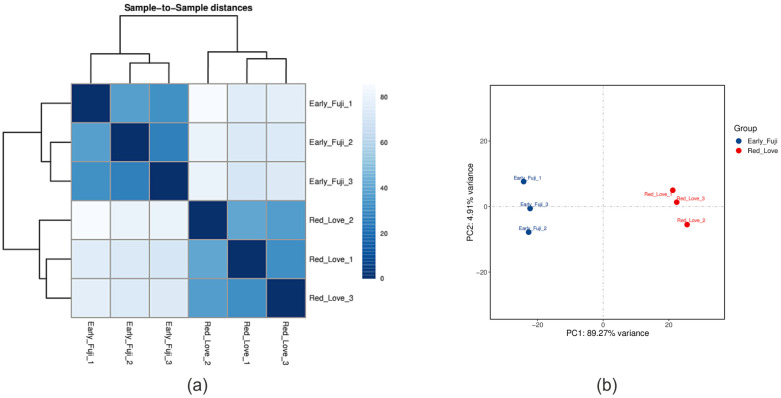
Heatmap of Pearson’s correlation (correlation coefficient between 1 and −1 (dark-blue squares represent the highest correlation value) (**a**), and PCA analysis between Red Love^®^ ‘General’ and ‘Early Fuji’ samples (**b**), both calculated based on gene expression FPKM values.

**Figure 2 ijms-25-01778-f002:**
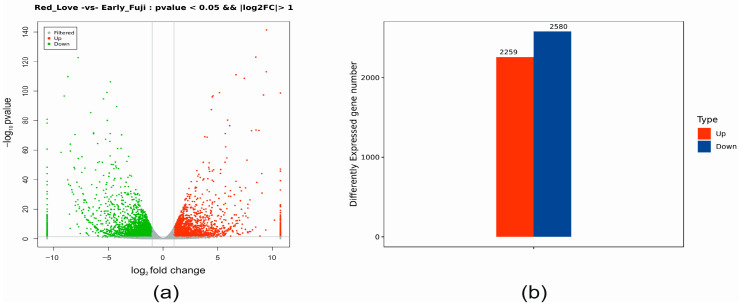
Summary of the differentially expressed genes (DEGs) identified in fruits of red-fleshed apple cultivar Red Love^®^ ‘General’, as compared to white-fleshed ‘Early Fuji’. (**a**) The volcano plot shows the filtered data extraction of genes up- vs. down-regulated in accordance with the *p*-value. (**b**) Numbers of DEGs detected in comparison.

**Figure 3 ijms-25-01778-f003:**
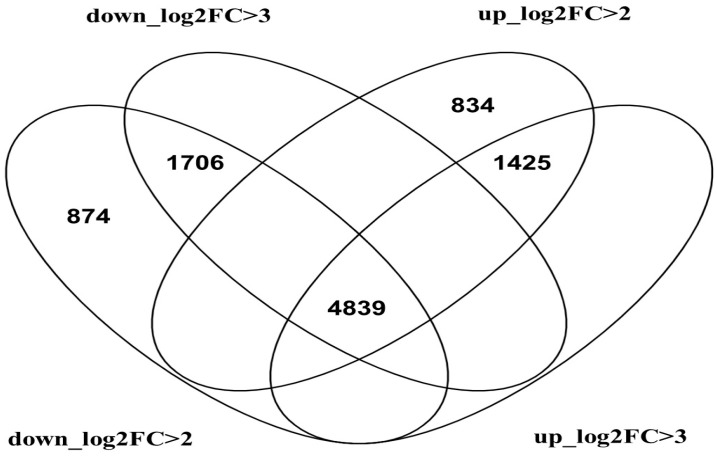
Venn diagram representing the numbers of differentially expressed genes (DEGs) at different levels of expression folds: log2 fold-change > 2 (log2FC > 2) and log2 fold-change > 3 (log2FC > 3).

**Figure 4 ijms-25-01778-f004:**
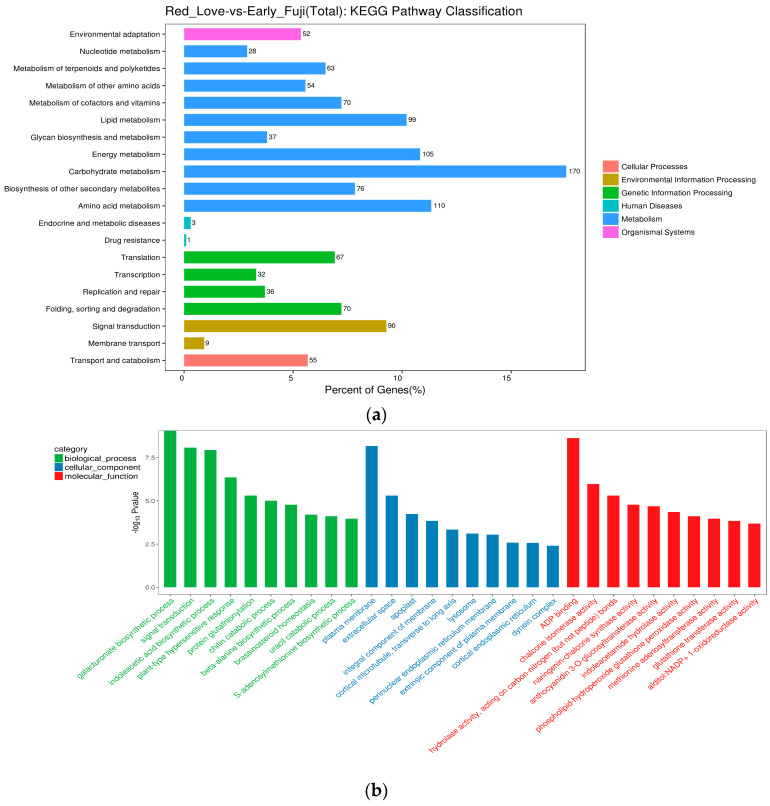
Summary of the DEG functional enrichment analysis. (**a**) Kyoto Encyclopedia of Genes and Genomes (KEGG) pathway classification of DEGs identified in red-fleshed fruits of Red Love^®^ ‘General’ in comparison with white-fleshed ‘Early Fuji’. (**b**) GO term classification of top genes up-regulated in red-flesh fruits of Red Love^®^ ‘General’.

**Figure 5 ijms-25-01778-f005:**
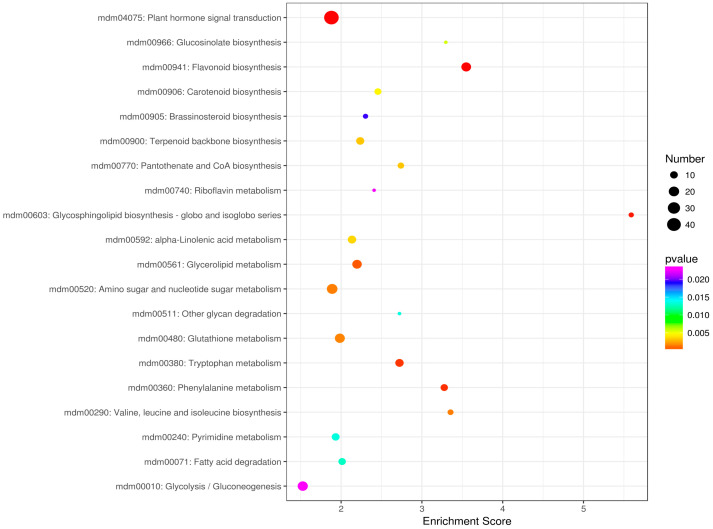
Classification of genes from the top 20 selected KEGG enrichments based on fold-change significance value.

**Figure 6 ijms-25-01778-f006:**
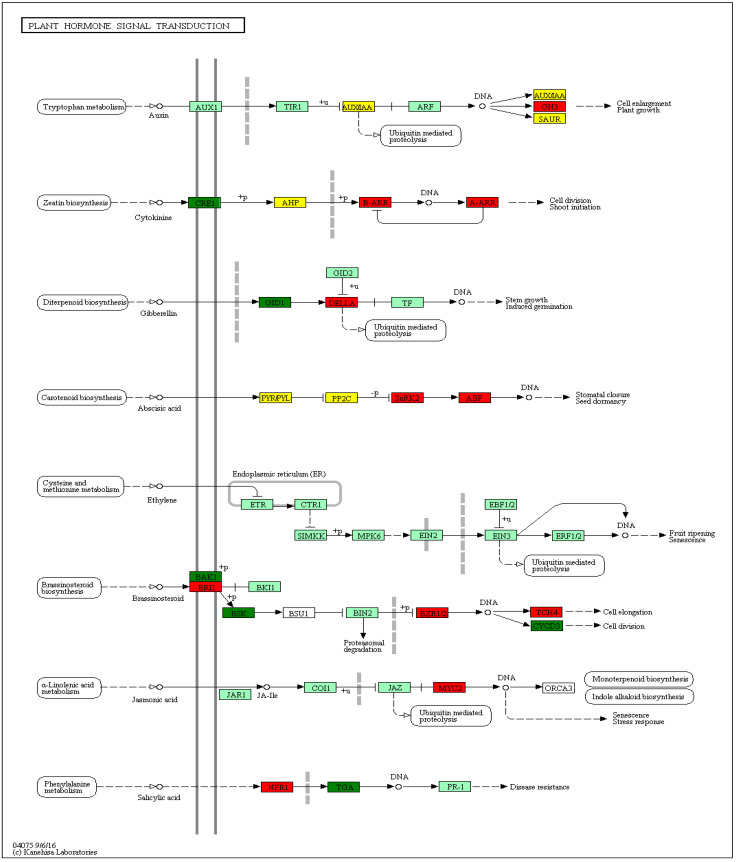
KEGG plant hormone signal transduction pathway with differentially expressed genes in fruits of red-fleshed Red Love^®^ ‘General’. Red-marked genes are up-regulated, green-marked genes are down-regulated, and yellow-marked genes are up- or down-regulated.

**Figure 7 ijms-25-01778-f007:**
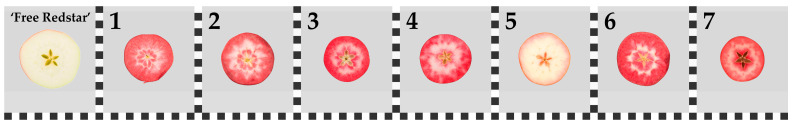
Visualization of apple fruit-flesh coloration; ‘Free Redstar’—fruit-flesh control; 1—‘Trinity’, 2—‘Alex Red’, 3—Red Love^®^ ‘Era’, 4—Red Love^®^ ‘Circe’, 5—‘Roxana’, 6—*M. sieviersii* f. *niedzwetzkyana*, 7—Red Love^®^ ‘Sirena’; white and black ruler (with default 1 × 1 cm dimension squares) illustrates fruit size.

**Figure 8 ijms-25-01778-f008:**
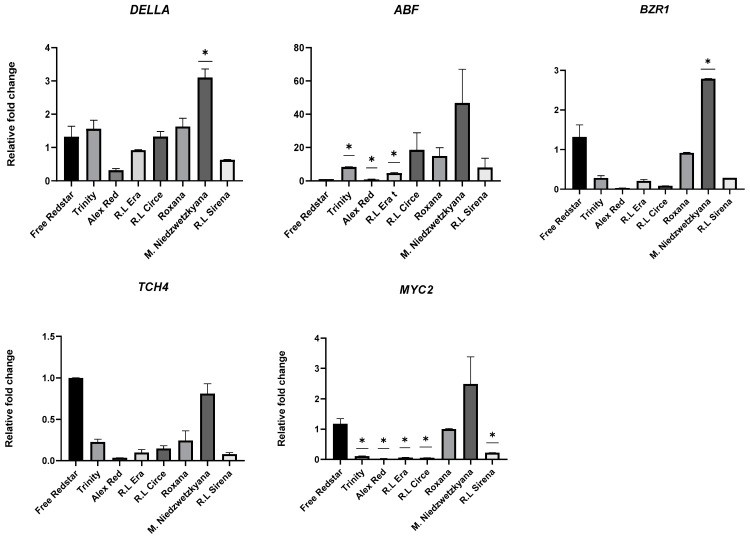
Validation of differentially expressed genes for immature fruits of apple cultivars with different flesh colors (only significant changes in gene expression were shown). Diagrams present average relative gene expression data with standard error of the mean (±SEM) compared to white flesh ‘Free Redstar’ and *t*-test significance calculation level *p* < 0.05 *, normalized to the *ACTIN* gene (showing stable expression in the experiment layout). The relative expression of genes of interest was calculated using mathematical equation 2^−ΔΔCT^ (RotorGene 6000 Series software 1.7) and visualized with GraphPad Prism10.0.3 software.

**Figure 9 ijms-25-01778-f009:**
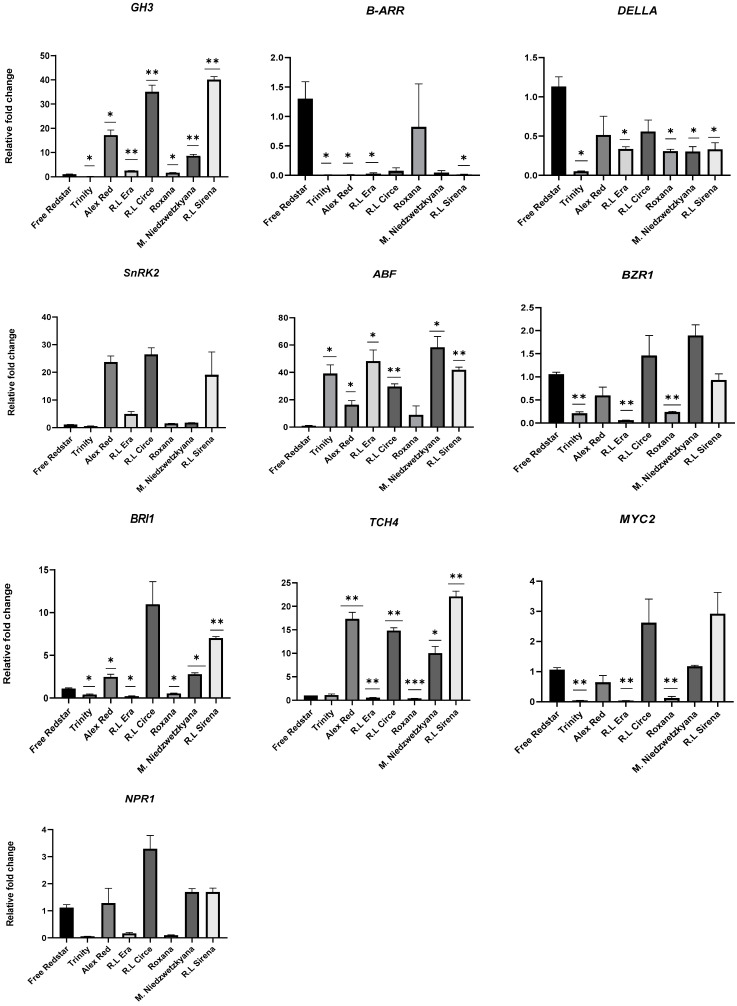
Validation of differentially expressed genes in ripe fruits of the apple cultivars analyzed. Diagrams present an average relative gene expression data with standard error of the mean (±SEM) compared to white flesh ‘Free Redstar’ and *t*-test significance calculation level *p* < 0.05 *, 0.01 **, 0.001 ***, normalized to the *ACTIN* gene (showing stable expression in the experiment layout). The relative expression of genes of interest was calculated using mathematical equation 2^−ΔΔCT^ (RotorGene 6000 Series software 1.7) and visualized with GraphPad Prism10.0.3 software.

**Figure 10 ijms-25-01778-f010:**
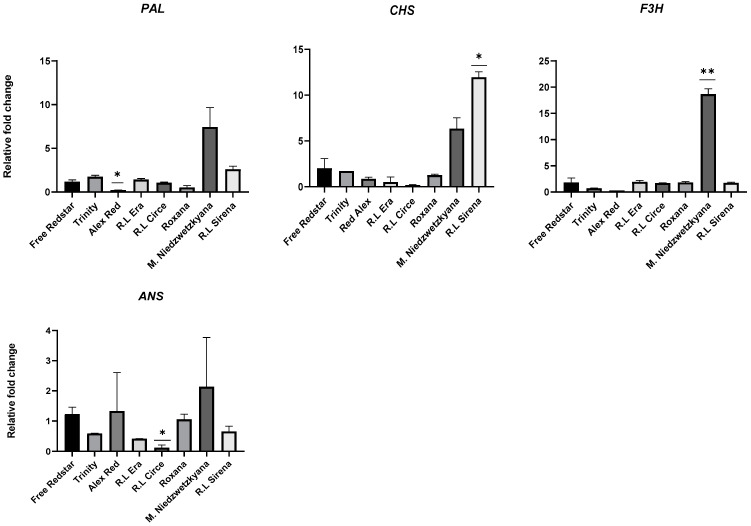
Expression profiles of structural genes of the anthocyanin biosynthesis pathway, evaluated in immature fruits of apple cultivars. Diagrams present average relative gene expression data with standard error of the mean (±SEM) compared to white flesh ‘Free Redstar’ and *t*-test significance calculation level *p* < 0.05 *, 0.01 ** normalized to the *ACTIN* gene (showing stable expression in the experiment layout). The relative expression of genes of interest was calculated using mathematical equation 2^−ΔΔCT^ (RotorGene 6000 Series software 1.7) and visualized with GraphPad Prism10.0.3 software.

**Figure 11 ijms-25-01778-f011:**
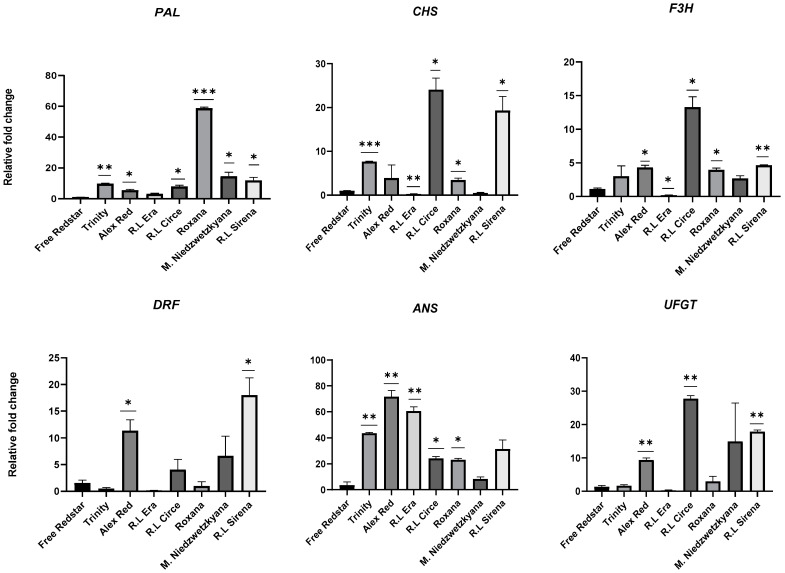
Expression profiles of structural genes of the anthocyanin biosynthesis pathway, evaluated in ripe fruits of apple cultivars. Diagrams present average relative gene expression data with standard error of the mean (±SEM) compared to white flesh ‘Free Redstar’ and *t*-test significance calculation level *p* < 0.05 *, 0.01, **, 0.001 ***, normalized to the *ACTIN* gene (showing stable expression in the experiment layout). The relative expression of genes of interest was calculated using mathematical equation 2^−ΔΔCT^ (RotorGene 6000 Series software 1.7) and visualized with GraphPad Prism10.0.3 software.

**Figure 12 ijms-25-01778-f012:**
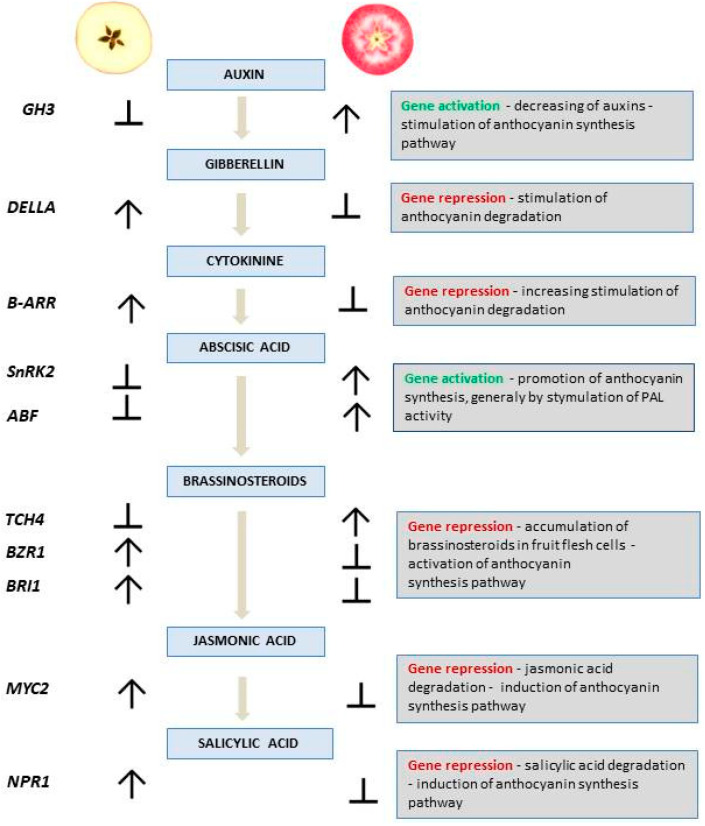
Simulated scheme of the mechanism for anthocyanin synthesis regulation predicted by the activity modulation of genes from plant hormones and transduction pathways. Common arrows and blunt arrows show gene regulation (positive and negative regulatory mechanisms, respectively) in response to the hormone signaling in red-/white-flesh apple phenotypes. The right column shows the description of the putative mechanisms for anthocyanin accumulation in red-fleshed apples.

**Table 1 ijms-25-01778-t001:** Characterization of the DEGs involved in hormonal signaling, up-regulated in red-fleshed fruits of RedLove^®^ ‘General’.

Gene Name	Locus	Hormone Signaling Pathway	Cellular Localization and Gene Function	FoldChange (FC)
*GH3*	LOC103436425	Auxin	probable indole-3-acetic acid-amido synthetase GH3.1; auxin responsive GH3 gene family	24.2
*B-ARR*	LOC103400015	Cytokinin	two-component response regulator ARR1-like isoform X1; regulator ARR-B family; response_reg Myb_DNA-binding	4.1
*DELLA*	LOC103406747	Gibberellin	DELLA protein GAI-like; DELLA protein	2.6
*SnRK2*	LOC103429475	ABA	serine/threonine-protein kinase SRK2; kinase PK_Tyr_Ser-Thr Choline_kinase PPP1R21_C; cytoplasm, signaling pathway kinase family.	2.7
*ABF*	LOC103446587	ABA	ABSCISIC ACID-INSENSITIVE 5-like protein 2; ABA responsive element binding factor.	20.6
*BRI1*	LOC103410973	Brassinosteroids	plasma membrane, receptor S160/brassinosteroid insensitive protein 1	2.2
*BZR1/2*	LOC103440434	Brassinosteroids	BES1/BZR1 homolog protein 2-like; brassinosteroid resistant 1/2	5.2
*TCH4*	LOC103409272	Brassinosteroids	probable xyloglucan endotransglucosylase/hydrolase protein 23 precursor; cell wall xyloglucan:xyloglucosyl transferase TCH4;	5.7
*MYC2*	LOC103404780	Jasmonic acid	transcription factor; bHLH-MYC	2.1
*NPR1*	LOC103454562	Salicylic acid	protein ubiquitination, BTB/POZ domain and ankyrin repeat-containing regulatory protein NPR1	2.9

**Table 2 ijms-25-01778-t002:** Correlation matrix calculated between the activity of genes from plant signal and hormone transduction and anthocyanin biosynthesis pathways, calculated by Pearson rank evaluation with significances of *p* > 0.05 (*), 0.01 (**), 0.001 (***) and 0.0001 (****), evaluated for the flesh of immature (**a**) and ripe (**b**) apple fruits.

**(a)**										
	** *ABF* **	** *B-ARR* **	** *BZR1* **	** *NPR1* **	** *DELLA* **	** *GH3* **	** *MYC2* **	** *SnRK2* **	** *BRI1* **	** *TCH4* **
*ANS*					**					
*CHS*										
*DRF*		****		***		**		**	****	
*F3H*		****	*	****		*		**	****	
*PAL*			*		*					
*UFGT*		**		*		****		***	***	*
**(b)**										
	** *ABF* **	** *B-ARR* **	** *BZR1* **	** *NPR1* **	** *DELLA* **	** *GH3* **	** *MYC2* **	** *SnRK2* **	** *BRI1* **	** *TCH4* **
*ANS*										
*CHS*		*				*		*		*
*DFR*										
*F3H*										
*PAL*		**				**		**		
*UFGT* ^1^	***	****	**	*	**	****	*	***	****	***

^1^ The significant correlation between genes from the plant hormone transduction pathway and *UFGT* (anthocyanin biosynthesis gene) is underlined in ripe, red-fleshed fruits.

**Table 3 ijms-25-01778-t003:** Names and sequences of primers used to study expression profiles of genes involved in anthocyanin biosynthesis in apple fruits.

	Gene Abbreviation	Oligo 5′	Oligo 3′	Reference
Differentially expressed genes	*TCH4*	CTCAACTGGGGAACCCTACA	GGCATTCCAAAAAGTTGCAT	This study—revealed in RNA-seq
	*BZR1*	TAGTCCGTCGTCTTCGTCCT	GAGACGGCGTAAAATGGGTA	
	*BRI1*	GCTTTGGACCACCTTGACAT	CACAAGCTCTGCACACGAAT	
	*MYC2*	TGTTTGGGCTGCAGACTATG	TCCTTCATTTCCATGGTGG	
	*NPR1*	GCCTTGAGCTCGTACAGTCC	AGACCCCATTTGATGAGCTG	
	*GH3*	ACAGATCCTTCCCGCCTAGT	AATTGGTGCGCCATAGGTAG	
	*B-ARR*	ACTTGCTTCGCCAAAAGAAA	TGCCATATATTGCGCAGTTC	
	*DELLA*	TAGTGACGGTTGTGGAGCAG	CTCCACTTGCTTAGCGGTTC	
	*SnRK2*	GGCGAATCCTTACTGTACGC	GTCTATGCTCTGGGCTGGAG	
	*ABF*	ACAACGGTCACCATCAACAA	CTGACGTCCTCTTCCCTCAC	
Structural genes	*ANS*	CAATTTGGCCTCAAACACCT	TGAGCTTCAACACCAAGTGC	Kondo et al. [56]
	*PAL*	CGGAAACTTGGACTCGGTAA	GATGGAGCCTCTTGCTTGTC	
	*DFR*	GAGTCCGAATCCGTTTGTGTCA	ATGTTTGTGGGGGCTGTCGATG	
	*UFGT*	TCCCTTTCACTAGCCATGCAAG	GTGGAGGATGGAGTTTTTACC	
	*F3H*	GGTGAACTCAAACAGCAGCA	CCACTTTGGCTTTCTCCAAG	
	*CHS*	ACCCACTTGGTCTTTTGCAC	ACTAGGCCCTCGGAAGGTAA	
Reference	*ACTIN*	GACTGTGAAACTGCGAATGGCTCA	CATGAATCATCAGAGCAACGGGCA	Xu et al. [30]

## Data Availability

The RNA-seq datasets associated with this study were submitted into the NCBI Sequence Read Archive (SRA) database under Bioproject ID: PRJNA1067510.

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
