# Peer review of "Transcriptome Analysis of White- and Red-Fleshed Apple Fruits Uncovered Novel Genes Related to the Regulation of Anthocyanin Biosynthesis"

_ijms, 2024, doi:10.3390/ijms25031778_

Round 1

Reviewer 1 Report

Comments and Suggestions for Authors

The presented results are interesting, original and potentially valuable for molecular selection of red-fleshed apples. The preliminary findings based on the correlation analyses should be further examined in the nearby future. 

Please replace the citations 1 and 2 by more appropriate ones.

Please replace commas in decimal numbers by dots.

Fig.1  Please enhance the heatmap resolution. Please clarify color scale. The figure is confusing. Part b) represents PCA, not hierarchical clustering. Please correct the figure caption and add Y axis label.

Fig.2b  Please consider Venn diagram instead of bar plot.

As the differences between up-and downregulated genes are not significant please consider omitting GO classification of differentially expressed genes and leave just summary.

Moreover, it is worth to think about the lower FC values. I would suggest to calculate fold changes by another method, then compare the results to make them more reliable and also to confirm FC cut-off as sufficient.

Comments on the Quality of English Language

Please check the potential typing errors, e.g. "weather" at line 527, "signallig" at line 226, etc. 

Author Response

26.01.2024

Keller-Przybylkowicz Sylwia

The National Institute of Horticultural Research;

Konstytucji 3-go Maja, 96-100 Skierniewice, Poland

International Journal of Molecular Science

Dear Managing Editor

Ms. Kaitlyn Wu

[Cover Letter]

We would like to thank editorial board and reviewers for every remark that helped us to improve the current version of the manuscript.

We appreciate you and the reviewers for your precious time in reviewing our paper and providing valuable comments.

We have carefully considered the comments and tried our best to address every one of them. We hope the manuscript after careful revisions meet your high standards. The authors welcome further constructive comments if any.

Below we provide the point-by-point responses. All modifications in the manuscript have been highlighted in current version accordingly to the track changes of word document.

Sincerely,

Sylwia Keller-Przybylkowicz,

Sylwia.Keller@inhort.pl

The National Institute of Horticultural Research

Konstytucji 3-go Maja 1/3

96-100 Skierniewice

Poland

Author’s responses to the comments of the reviewer #1 of the manuscript entitled “Expression profiling uncovers novel genes associated with plant hormone signal transduction involved in regulation of anthocyanin biosynthesis in red-fleshed apple cultivars”, submitted to International Journal of Molecular Science under the tracking number: ijms-2839210.

  1. The presented results are interesting, original and potentially valuable for molecular selection of red-fleshed apples. The preliminary findings based on the correlation analyses should be further examined in the nearby future.

Author’s response: Yes, the correlations are very interesting and it will be followed by us in GWAS study. The selected genes and their association with apple fruit flesh pigmentation will be examined in the near future, and the result will be presented in subsequent papers.

  1. Please replace the citations 1 and 2 by more appropriate ones.

Author’s response: The remark has been accepted.

The citation: 

Velasco, R.; Zharkikh, A.; Affourtit, J.; Dhingra, A.; Cestaro, A.; Kalyanaraman, A.; Fontana, P.; Bhatnagar, S.K.; Troggio, M.; Pruss, D.; et al. The genome of the domesticated apple (Malus x domestica Borkh.). Nat. Genet. 2010, 42, 833–839. https://doi.org/10.1038/ng.654

Was replaced into;

Liu, W.; Chen, Z.; Jiang, S.; Wang, Y.; Fang, H.; Zhang, Z.; Chen, X.; Wang, N. Research Progress on Genetic Basis of Fruit Quality Traits in Apple (Malus × domestica). Front Plant Sci. 2022, 13, 918202. https://doi.org/10.3389/fpls.2022.918202

The citation: 

Wang, N.; Chen, X. Genetics and genomics of fruit color development in apple. In The Apple Genome; Korban, S.S., Ed.; Springer International Publishing: Cham, Switzerland, 2021; pp. 271–295. https//:doi.org/10.1007/978-3-030-74682-7_13

Was replaced into;

Chen, Z.; Yu, L.; Liu, W.; Zhang, J.; Wang, N.; Chen, X. Research progress of fruit color development in apple (Malus domestica Borkh.). Plant Physiol. Biochem. 2021, 162, 267–279. https://doi.org/10.1016/j.plaphy.2021.02.033

  1. Please replace commas in decimal numbers by dots.

Author’s response: The remark has been accepted, the commas in the table 3 were replaced by dots.

  1. Fig.1 Please enhance the heatmap resolution. Please clarify color scale. The figure is confusing. Part b) represents PCA, not hierarchical clustering. Please correct the figure caption and add Y axis label.

Author’s response: Remark has been accepted. The heatmap was presented and now is clarified, the PCA plot was also improved in regard to image quality and the Y axis label was added to the chart.

The figure description was adjusted accordingly and now it is:

Figure 1. Heatmap of Pearson’s correlation (correlation coefficient between 1, -1 (dark blue squares represent the highest correlation value) (a), and PCA analysis between Red Love® ‘General’ and ‘Early Fuji’ samples (b) both calculated based on gene expression FPKM values.

  1. Fig.2b Please consider Venn diagram instead of bar plot.

Authors’ response: Remark has been accepted. Thank You for the precious suggestion that helped us to improve the manuscript. The Venn diagram has been added to the manuscript as Figure 3.

Accordingly to the Venn graph presentation, in the paragraph 2.1 transcription profiling, the phrase:

Up on the filtered data, a Venn diagram was prepared to indicate the number of up-regulated genes with the log2 fold-change >2 (2 259 DEGs) and log2 fold-change >3 (1 425 DEGs), as well as down-regulated genes with the log2 fold-change >2 (2 580 DEG’s) and with the log2 fold-change >3 (1 706 DEGs). According to the log 2 fold-change >2 - 874 down and 834 up regulated unique genes were revealed (Figure 3).

-was added.

  1. As the differences between up-and downregulated genes are not significant please consider omitting GO classification of differentially expressed genes and leave just summary.

Author’s response: Remark has been accepted and Figure 3 in the previous shape is now removed and replaced by Venn diagram.

  1. Moreover, it is worth to think about the lower FC values. I would suggest to calculate fold changes by another method, then compare the results to make them more reliable and also to confirm FC cut-off as sufficient.

Author’s response: The authors accept this suggestion to use other methods of calculating FC values in subsequent publications devoted to transcriptome analysis. Also Venn diagram, added in the results paragraph, shows the relations between most differentially expressed genes.

  1. Comments on the Quality of English Language

Please check the potential typing errors, e.g. "weather" at line 527, "signallig" at line 226, etc.

Author’s response: The remark has been accepted. The language was improved

Acknowledgments were added to the manuscript:

Acknowledgments: The manuscript presents the results carried out as part of the 'Double Hundred Talents Exchange' project in cooperation agreement between The National Institute of Horticultural Research, Skierniewice, Poland, and the Yantai Academy of Agricultural Science, Yantai China.

Reviewer 2 Report

Comments and Suggestions for Authors

15.1.2024

The manuscript entitled “Expression profiling uncovers novel genes associated with plant hormone signal transduction involved in regulation of anthocyanin biosynthesis in red-fleshed apple cultivars” was reviewed.

The manuscript delivers novel DGEs date related to anthocyanin biosynthesis in red-fleshed apples. However, few important issues are pending, including weak language and discussion. Moreover, RNAseq data should be deposited to the Genbank to serve the scientific community. Therefore, I do not recommend the publication of the manuscript in “International Journal of Molecular Sciences” unless these comments and corrections are applied (please see below).

1. General:

- Title: the title is too long! Please, shorten it and be precise.

- The Language is weak, please correct for common grammar mistakes and incorrect sentence phrasing. Please see comments related to the “Abstract” section below as an example for the entire manuscript.

- After first appearance in full, the genus “Malus” should be abbreviated into “M.” thereafter.

2. Abstracts:

- Line 15: remove "of the performance".

- Line 19: remove "so far".

- Line 20: replace "previously unrecognized" to "novel".

- Line 20: replace "involved in the constitution" with "related to".

- Line 23: replace "ten" with "ten potential".

- Lines 25-26: remove "In presented studies"

- Line 26: remove "their".

- Line 27: remove "Generally,".

- Line 27: replace ", the set of the novel genes," with "genes".

- Line 28: remove "the".

- Line 28: remove "preliminarily".

- Lines 29-30: remove "Their expression was not characterized in apple fruits so far."

- Line 30: replace "study, allowed to" with "results".

- Line 31: replace "of" with "between".

- Line 31: replace "directly" with "which is directly".

- Lines 32-33: remove "confirmed their crucial role of plant hormone regulation and"

3. Introduction:

- The Language is weak, please correct for common grammar mistakes and incorrect sentence phrasing. - Line 38: remove “worldwide” as it contradicts with “world temperate regions”.

- Lines 38-39: replace “ecological adaptability” with “economic value”.

- Line 39: replace “The global” with “The annual global”.

- Line 44: remove “of” from “increasing of”.

- Line 45: replace “by introduction into apple industry red-fleshed varieties,” with “by introducing red-flesh apples, which are”.

- Line 46: replace “having” with “and have”.

- Line 46: add “,” after “In addition”.

- Line 48: replace “wildly spread” with “intensive”.

- ….

4. Results:

- Line 127: very low-resolution figure, please replace with high quality images.

- Line 137: very low-resolution figure, please replace with high quality images.

- Line 137: it is recommended to add Venn diagram showing overlapping genes with different expression folds, e.g. 2-fold, 3-fold, … etc.

- Line 149: very low-resolution figure, please replace with high quality images.

- Line 261: you need to italicize all gene names!

- Line 261: as DGEs were calculated for all cultivars relative to ‘Free Redstar’, the later SHOULD have relative fold change of ONE! How comes it is ca. 1.2 for DELLA and BZR, while it is > 50 for ABF???

- Line 281: AGAIN, as DGEs were calculated for all cultivars relative to ‘Free Redstar’, the later SHOULD have relative fold change of ONE! How comes it is > 1 for several genes???

5. Discussion:

- Relatively weak. You presented huge and novel results, but you discussion is limited. Please cited more recent articles and expand your discussion.

- Line 387: instead of using the “strange” uncommon “RED PROHIBITITION SIGN”, please use common arrows and blunt arrows for “expression” and “repression”, respectively, as in “Figure 6”.

- Line 387: very low-resolution figure, please replace with high quality images.

6. Materials and Methods:

- Line 293: did you apply “DNase I Digestion” as it is optional in this kit? But it is a must for RNAseq!

- Line 509: the NGS of the RNAseq should be deposited as SRAs in the Genbank (NCBI) before accepting the publication to serve the scientific community. This is the norm with NGS articles.

- Line 536: replace “Cell” with “Cellular”.

- Lines 539-540: you are repeating the full names of CC, BP and MF as you did in line 536. Use only abbreviated form in the second time.

- Line 551: need to indicate the number of biological replicates used in RT-qPCR.

- Line 563: the normal used form of PCR primers is “capital letters”, please modify accordingly.

- Line 583: replace “gene-to-gen” with “gene-to-gene”.

- Line 583: replace “gens was” with “genes was”.

7. References:

- Cited articles are too old, only 12 out of 66 (18%) are published in the last five years. You need to cite more recent articles and remove some of the old or one-time cited articles! 

Comments on the Quality of English Language

- The Language is weak, please correct for common grammar mistakes and incorrect sentence phrasing. Please see comments related to the “Abstract” section as an example for the entire manuscript.

Author Response

26.01.2024

Keller-Przybylkowicz Sylwia

The National Institute of Horticultural Science;

Konstytucji 3-go Maja, 96-100 Skierniewice, Poland

International Journal of Molecular Science

Dear Managing Editor

Ms. Kaitlyn Wu

[Cover Letter]

We would like to thank editorial board and reviewers for every remark that helped us to improve the current version of the manuscript.

We appreciate you and the reviewers for your precious time in reviewing our paper and providing valuable comments.

We have carefully considered the comments and tried our best to address every one of them. We hope the manuscript after careful revisions meet your high standards. The authors welcome further constructive comments if any.

Below we provide the point-by-point responses. All modifications in the manuscript have been highlighted in current version accordingly to the track changes of word document.

Sincerely,

Sylwia Keller-Przybylkowicz,

Sylwia.Keller@inhort.pl

The National Institute of Horticultural Research

Konstytucji 3-go Maja 1/3

96-100 Skierniewice

Poland

Authors' response to the comments of the reviewer #2 of the manuscript entitled “Expression profiling uncovers novel genes associated with plant hormone signal transduction involved in regulation of anthocyanin biosynthesis in red-fleshed apple cultivars”, submitted to International Journal of Molecular Science under the tracking number: ijms-2839210.

  1. General:

- Title: the title is too long! Please, shorten it and be precise.

Authors’ response: Remark has been accepted. In accordance to the reviewer suggestion the title was shorten from “Expression profiling uncovers novel genes associated with plant hormone signal transduction involved in regulation of anthocyanin biosynthesis in red-fleshed apple cultivars” into “Transcriptome analysis of white and red-fleshed apple fruits uncovered novel genes related to the regulation of anthocyanin biosynthesis”.

- The Language is weak, please correct for common grammar mistakes and incorrect sentence phrasing. Please see comments related to the “Abstract” section below as an example for the entire manuscript.

Authors’ response: Remark has been accepted. The manuscript was revised according to the suggestion.

- After first appearance in full, the genus “Malus” should be abbreviated into “M.” thereafter.

Authors’ response: Remark has been accepted.

  1. Abstract:

- Line 15: remove "of the performance".

- Line 19: remove "so far".

- Line 20: replace "previously unrecognized" to "novel".

- Line 20: replace "involved in the constitution" with "related to".

- Line 23: replace "ten" with "ten potential".

- Lines 25-26: remove "In presented studies"

- Line 26: remove "their".

- Line 27: remove "Generally,".

- Line 27: replace ", the set of the novel genes," with "genes".

- Line 28: remove "the".

- Line 28: remove "preliminarily".

- Lines 29-30: remove "Their expression was not characterized in apple fruits so far."

- Line 30: replace "study, allowed to" with "results".

- Line 31: replace "of" with "between".

- Line 31: replace "directly" with "which is directly".

- Lines 32-33: remove "confirmed their crucial role of plant hormone regulation and"

Authors’ response: All the remarks have been accepted and the text has been adjusted accordingly.

  1. Introduction:

- The Language is weak, please correct for common grammar mistakes and incorrect sentence phrasing. - Line 38: remove “worldwide” as it contradicts with “world temperate regions”.

- Lines 38-39: replace “ecological adaptability” with “economic value”.

- Line 39: replace “The global” with “The annual global”.

- Line 44: remove “of” from “increasing of”.

- Line 45: replace “by introduction into apple industry red-fleshed varieties,” with “by introducing red-flesh apples, which are”.

- Line 46: replace “having” with “and have”.

- Line 46: add “,” after “In addition”.

- Line 48: replace “wildly spread” with “intensive”.

- ….

Authors’ response: All the remarks have been accepted and the text has been adjusted accordingly.

  1. Results:

- Line 127: very low-resolution figure, please replace with high quality images.

- Line 137: very low-resolution figure, please replace with high quality images.

Authors’ response: The images were improved in regard to resolution and quality.

- Line 137: it is recommended to add Venn diagram showing overlapping genes with different expression folds, e.g. 2-fold, 3-fold, … etc.

Authors’ response: Remark has been accepted. Thank You for the precious suggestion that helped us to improve the manuscript. The Venn diagram has been added to the manuscript as Figure 3.

Accordingly to the Venn graph presentation, in the paragraph 2.1 transcription profiling, the phrase:

Up on the filtered data, a Venn diagram was prepared to indicate the number of up-regulated genes with the log2 fold-change >2 (2 259 DEGs) and log2 fold-change >3 (1 425 DEGs), as well as down-regulated genes with the log2 fold-change >2 (2 580 DEG’s) and with the log2 fold-change >3 (1 706 DEGs). According to the log 2 fold-change >2 - 874 down and 834 up regulated unique genes were revealed (Figure 3).

- was added

Figure 3 in the previous shape is now removed (according to the reviewer #1 remark) and replaced by Venn diagram.

- Line 149: very low-resolution figure, please replace with high quality images.

Authors’ response: Remark has been accepted. The image has been replaced into high quality resolution.

- Line 261: you need to italicize all gene names!

Authors’ response: All the gene terms were presented in italic, including the expression profiles charts.

- Line 261: as DGEs were calculated for all cultivars relative to ‘Free Redstar’, the later SHOULD have relative fold change of ONE! How comes it is ca. 1.2 for DELLA and BZR, while it is > 50 for ABF???

- Line 281: AGAIN, as DGEs were calculated for all cultivars relative to ‘Free Redstar’, the later SHOULD have relative fold change of ONE! How comes it is > 1 for several genes???

Authors’ response: Remark has been accepted. Thank you very much for pointing it out.

The reviewer observed an inconsistency in the value of the 'Free Redstar' control, which should have been designated as 1. In presented research we obtained a relative fold change value close to 1 (i.e. 1.2, 0.8), resulting from rounding the averages. The tests included samples containing separate fruits replicates (three in each analysis), which were collected in the principal development stage of maturation. According to the fruit ripening physiology, the replicates weren’t identical; hence small deviations in the analysis of relative gene expression in the 'Free Redstar' control sample (white-fleshed) were detected.

The diagram for ABF gene expression profile was corrected according to the reviewer remarks.

The authors admit that there was an error in the data entered in the chart prepared for the ABF gene. Due to the very extensive scope of the work and many aspects touched upon in it, we did not avoid any written errors. A new correct chart has been introduced into the manuscript.

  1. Discussion:

- Relatively weak. You presented huge and novel results, but you discussion is limited. Please cited more recent articles and expand your discussion.

Authors’ response: According to the reviewer remark the discussion was improved based on the recent literature. Authors declare that there were limited novel literatures in regard the role of presented genes in apple species. In case of some aspects the more recent articles were added to the manuscript.

  1. Changes have been made in the sentence:

In conducted research, the newly studied DEGs such as DELLA (gibberellin pathway) and B-ARR (cytokinin pathway), showed significant down-regulation in the red-fleshed ripe fruits and up-regulation in white fruit flesh of ‘Free Redstar’.

Now it sounds:

In conducted research, the newly studied DEGs such as DELLA (gibberellin pathway and responsible for ubiquitin mediated proteolysis) and B-ARR (cytokinin pathway), showed significant down-regulation in the red-fleshed ripe fruits and up-regulation in white fruit flesh of ‘Free Redstar’.

  1. The sentences: In Arabidopsis and grape it was uncovered that DELLA gene also mediating environmental stimulation of anthocyanin biosynthesis [34]

and

In addition, as it was reported by Shi et al. cytokinin enhances sucrose-mediated anthocyanin pigmentation and, especially under plant stress response seems to play a negative role in anthocyanin accumulation [34].

were added in the Discussion paragraph.

- Line 387: instead of using the “strange” uncommon “RED PROHIBITITION SIGN”, please use common arrows and blunt arrows for “expression” and “repression”, respectively, as in “Figure 6”.

Authors’ response: Remark has been accepted, and the scheme was corrected according to the reviewer suggestion. The image was reconstructed accordingly and the arrows were changed into blunt and common ones.

- Line 387: very low-resolution figure, please replace with high quality images.

Authors’ response: Remark has been accepted. The resolution of the scheme image was improved.

  1. Materials and Methods:

- Line 293: did you apply “DNase I Digestion” as it is optional in this kit? But it is a must for RNAseq!

Authors’ response: The remark has been accepted. Thank you very much for pointing it out.

Authors admit that the step for DNaze I treatment was unfortunately omitted in the method description. Since there is an obligatory to apply this step in RNA samples preparation, this was also considered in our methodology.

In the paragraph 4.2. RNA extraction - the phrase:

‘The samples were treated with DNase I using Turbo DNA-free kit (Thermo Fisher Scientific, USA) according to the manufacturer's protocol.’  - was added.

- Line 509: the NGS of the RNAseq should be deposited as SRAs in the Genbank (NCBI) before accepting the publication to serve the scientific community. This is the norm with NGS articles.

Authors’ response: Thank You very much for pointing it out. Remark has been accepted. The sequence from RNAseq experiment was deposited in the NCBI GenBank.

In the paragraph 4.3 RNAseq analysis we have added the information about data submission:

The sentence was added to the manuscript body:

The RNA-seq datasets associated with this study were submitted into the National Center for Biotechnology Information (NCBI, Bethesda, MD, USA) Sequence Read Archive (SRA) database; under the accession:  SUB14156737, with Bioproject ID: PRJNA1067510.’

The data will be released upon acceptance of the manuscript.

- Line 536: replace “Cell” with “Cellular”.

Authors’ response: The remark has been accepted.

- Lines 539-540: you are repeating the full names of CC, BP and MF as you did in line 536. Use only abbreviated form in the second time.

Authors’ response: The remark has been accepted.

- Line 551: need to indicate the number of biological replicates used in RT-qPCR.

Authors’ response: The remark has been accepted. Thank you very much for pointing it out.

The number of replicates in RT-qPCR was added according to the reviewer suggestions. We revised the paragraphs as follows:

4.1. Plant material

‘Tissue samples (apple flesh from 3 fruits – biological replicates) were collected, from the apple trees grown in the experimental orchard maintained at the National Institute of Horticultural Research in Skierniewice, Poland.’

4.2. RNA extraction

‘For RNA-seq experiment, total RNA from 3 independent fruits of Red Love® ‘General’ and ‘Early Fuji’ was isolated using MINIBEST Plant RNA Extraction Kit (TaKaRa, China) according to manufacturer’s protocol.’

‘For RT-qPCR total RNA was isolated using method described by Zeng and Yang [60] with minor modifications. The procedure was started with sample incubation for 20 minutes in CTAB buffer heated to 65℃. After centrifugation in a mixture of chloroform and isoamyl alcohol (24:1, v/v) RNA (from 3 fruits replicates) was resolved from the upper solution fraction and precipitated with 10 M lithium chloride overnight at 4°C.’

- Line 563: the normal used form of PCR primers is “capital letters”, please modify accordingly.

Authors’ response: The remark has been accepted.

- Line 583: replace “gene-to-gen” with “gene-to-gene”.

Authors’ response: The remark has been accepted.

- Line 583: replace “gens was” with “genes was”.

Authors’ response: The remark has been accepted.

  1. References:

- Cited articles are too old, only 12 out of 66 (18%) are published in the last five years. You need to cite more recent articles and remove some of the old or one-time cited articles!

Authors’ response: According to the reviewer remark authors have added, replaced or removed reference positions. The more recent literature is now cited. This resulted in changes in the literature numbering, and below there is an explanation what changes has been made. All the remarks were revised in the manuscript.

References discarded:

  1. Zhou, Z.Q.; Li, Y.N. The RAPD evidence for the phylogenetic relationship of the closely related species of cultivated apple. Genet .Resour. Crop Evol.2000, 47, 353–357. https://doi.org/10.1023/A:1008740819941
  2. Pelletier, M.K.; Shirley, B.W. Analysis of flavanone 3-hydroxylase in Arabidopsis seedlings. Coordinate regulation with chalcone synthase and chalcone isomerase. Plant Physiol.1996, 111, 339–345. https://doi.org/10.1104/pp.111.1.339
  3. Dick-Pérez, M.; Zhang, Y.; Hayes, J.; Salazar, A.; Zabotina, O.A.; Hong, M. Structure and interactions of plant cell-wall polysaccharides by two- and three-dimensional magic-angle-spinning solid-state NMR. Biochemistry 2011, 50, 989–1000. https://doi.org/10.1021/bi101795q
  4. Honda, C.; Kotoda, N.; Wada, M.; Kondo, S.; Kobayashi, S.; Soejima, J.; Zhang, Z.; Tsuba, T.; Moriguchi, T. Anthocyanin biosynthetic genes are coordinately expressed during red coloration in apple skin. Plant Physiol. Biochem.2002, 40, 955–962. https://doi.org/10.1016/S0981-9428(02)01454-7
  5. Kondo, S.; Hiraoka, K.; Kobayashi, S.; Honda, C.; Terahara, N. Changes in the expression of anthocyanin biosynthetic genes during apple development. J. Am. Soc. Hortic. Sci.2002, 127, 971–997. https://doi.org/10.21273/JASHS.127.6.971
  6. Koes, R.; Verweij, W.; Quattrocchio, F. Flavonoids: a colorful model for the regulation and evolution of biochemical pathways. Trends Plant Sci.2005, 10, 236–242. https://doi.org/10.1016/j.tplants.2005.03.002
  7. Espley, R.V.; Brendolise, C.; Chagné, D.; Kutty-Amma, S.; Green, S.; Volz, R.; Putterill, J.; Schouten, H.J.; Gardiner, S.E.; Hellens, R.P.; Allan, A.C. Multiple repeats of a promoter segment causes transcription factor autoregulation in red apples. Plant Cell 2009, 21, 168–183. https://doi.org/10.1105/tpc.108.059329
  8. Espley, R.V.; Bovy, A.; Bava, C.; Jaeger, S.R.; Tomes, S.; Norling, C.; Crawford, J.; Rowan, D.; McGhie, T.K.; Brendolise, C.; Putterill, J.; Schouten, H.J.; Hellens, R.P.; Allan, A.C. Analysis of genetically modified red-fleshed apples reveals effects on growth and consumer attributes. Plant Biotechnol. J.2013, 11, 408–419. https://doi.org/10.1111/pbi.12017
  9. Ireland, H.S.; Guillen, F.; Bowen, J.H.; Tacken, E.; Putterill, J.; Schaffer, R.J.; Johnston, J.W. Mining the apple genome reveals a family of nine ethylene receptor genes. Postharvest Biol. Technol.2012, 72, 42–46. https://doi.org/10.1016/j.postharvbio.2012.05.003
  10. Ye, H.; Li, L.; Yin, Y. Recent advances in the regulation of brassinosteroid signaling and biosynthesis pathways. J.Integr. Plant Biol.2011, 53, 455–468. https://doi.org/10.1111/j.1744-7909.2011.01046.x
  11. Deikman, J.; Hammer, P.E. Induction of Anthocyanin Accumulation by Cytokinins in Arabidopsis thaliana.Plant Physiol.1995, 108, 47–57. https://doi.org/10.1104/pp.108.1.47

References exchanged:

  1. Velasco, R.; Zharkikh, A.; Affourtit, J.; Dhingra, A.; Cestaro, A.; Kalyanaraman, A.; Fontana, P.; Bhatnagar, S.K.; Troggio, M.; Pruss, D.; et al. The genome of the domesticated apple (Malus x domestica Borkh.).Nat. Genet.2010, 42, 833–839. https://doi.org/10.1038/ng.654

Into:

Liu, W.; Chen, Z.; Jiang, S.; Wang, Y.; Fang, H.; Zhang, Z.; Chen, X.; Wang, N. Research Progress on Genetic Basis of Fruit Quality Traits in Apple (Malus × domestica). Front Plant Sci. 2022, 13, 918202. https://doi.org/10.3389/fpls.2022.918202

  1. Wang, N.; Chen, X. Genetics and genomics of fruit color development in apple. In The Apple Genome; Korban, S.S., Ed.; Springer International Publishing: Cham, Switzerland, 2021; pp. 271–295. https//:doi.org/10.1007/978-3-030-74682-7_13

Into

Chen, Z.; Yu, L.; Liu, W.; Zhang, J.; Wang, N.; Chen, X. Research progress of fruit color development in apple (Malus domestica Borkh.). Plant Physiol. Biochem. 2021, 162, 267–279. https://doi.org/10.1016/j.plaphy.2021.02.033

  1. Van Nocker, S.; Berry, G.; Najdowski, J.; Michelutti, R.; Luffman, M.; Forsline, P.; Alsmairat, N.; Beaudry, R.; Nair, M.G.; Ordidge, M. Genetic diversity of red-fleshed apples (Malus). Euphytica2012, 185, 281–293. https://doi.org/10.1007/s10681-011-0579-7

Into:

Juhart, J.; Medic, A.; Veberic, R.; Hudina, M.; Jakopič, J.; Stampar, F. Phytochemical Composition of Red-Fleshed Apple Cultivar ‘Baya Marisa’ Compared to Traditional, White-Fleshed Apple Cultivar ‘Golden Delicious’. Horticulturae. 2022, 8, 811. https://doi.org/10.3390/horticulturae8090811

  1. Maliepaard, C.; Alston, F.H.; Van Arkel, G.; Brown, L.M.; Chevreau, E.; Dunemann, F.; Evans, K.M.; Gardiner, S.; Guilford, P.; Van Heusden, A.W.; Janse, J.; Laurens, F.; Lynn, J.R.; Manganaris, A.G.; den Nijs, A.P.M.; Periam, N.; Rikkerink, E.; Roche, P.; Ryder, C.; Sansavini, S.; Schmidt, H.; Tartarini, S.; Verhaegh, J.J.; Vrielink-van Ginkel, M.; King, G.J. Aligning male and female linkage maps of apple (Malus pumila Mill.) using multi-allelic markers. Theor. Appl. Genet.1998, 97, 60–73. https://doi.org/10.1007/s001220050867

Into:

Chengquan, Y.; Guangya, S.; Tao, W.; Baiquan, M.; Cuiying, Li.; Pengmin, Li.; Yangjun, Z.;  Lingfei, X.; Fengwang, M. Linkage Map and QTL Mapping of Red Flesh Locus in Apple Using A R1R1×R6R6 Population. Hortic. Plant J. 2020, 7, 393-400. https://doi.org/10.1016/j.hpj.2020.12.008

  1. Hertog, M.G.; Feskens, E.J.; Hollman, P.C.; Katan, M.B.; Kromhout, D. Dietary antioxidant flavonoids and risk of coronary heart disease: the Zutphen Elderly Study. Lancet 1993, 342, 1007–1011. https://doi.org/10.1016/0140-6736(93)92876-U

Into:

Yuste, S.; Ludwig, I.A.; Romero, M.P.; Piñol-Felis, C.; Catalán, Ú.; Pedret, A.; Valls, R.M.; Fernández-Castillejo, S.; Motilva, M.J.; Macià, A.; Rubió, L. Metabolic Fate and Cardiometabolic Effects of Phenolic Compounds from Red-Fleshed Apple in Hypercholesterolemic Rats: A Comparative Study with Common White-Fleshed Apple. The AppleCOR Study. Mol. Nutr. Food. Res. 2021, 65, 2001225. https://doi.org/10.1002/mnfr.202001225

  1. Knekt, P.; Kumpulainen, J.; Järvinen, R.; Rissanen, H.; Heliövaara, M.; Reunanen, A.; Hakulinen, T.; Aromaa, A. Flavonoid intake and risk of chronic diseases. Am. J. Clin. Nutr.2002, 76, 560–568. https://doi.org/10.1093/ajcn/76.3.560

Into:

Bhat, F.M.; Verma, R.; Chauhan, N. Apple and Apple Products: Properties and Health Benefits. ES. J. Nutr. Health. 2023, 3, 1017.

  1. Takos, A.M.; Jaffé, F.W.; Jacob, S.R.; Bogs, J.; Robinson, S.P.; Walker, A.R. Light-induced expression of a MYB gene regulates anthocyanin biosynthesis in red apples. Plant Physiol.2006, 142, 1216–1232. https://doi.org/10.1104/pp.106.088104

Into:

Shi, C.; Liu, L.; Wei, Z.; Liu, J.; Li, M.; Yan, Z.; Gao, D. Anthocyanin Accumulation and Molecular Analysis of Correlated Genes by Metabolomics and Transcriptomics in Sister Line Apple Cultivars. Life 2022, 12, 1246. https://doi.org/10.3390/life12081246

  1. Treutter, D. Biosynthesis of phenolic compounds and its regulation in apple. Plant Growth Regul.2001, 34, 71–89. https://doi.org/10.1023/A:1013378702940

Into:

Huang, Y.; Li, W.; Jiao, S.; Huang, J.; Chen, B. MdMYB66 Is Associated with Anthocyanin Biosynthesis via the Activation of the MdF3H Promoter in the Fruit Skin of an Apple Bud Mutant. Int. J. Mol. Sci. 2023, 24, 16871. https://doi.org/10.3390/ijms242316871

  1. Hichri, I.; Barrieu, F.; Bogs, J.; Kappel, C.; Delrot, S.; Lauvergeat, V. Recent advances in the transcriptional regulation of the flavonoid biosynthetic pathway. J. Exp. Bot.2011, 62, 2465–2483. https://doi.org/10.1093/jxb/erq442

Into:

Liu, H.; Liu, Z.; Wu, Y.; Zheng, L.; Zhang, G. Regulatory Mechanisms of Anthocyanin Biosynthesis in Apple and Pear. Int. J. Mol. Sci. 2021, 22, 8441. https://doi.org/10.3390/ijms22168441

  1. Symons, G.M.; Davies, C.; Shavrukov, Y.; Dry, I.B.; Reid, J.B.; Thomas, M.R. Grapes on steroids. Brassinosteroids are involved in grape berry ripening. Plant Physiol.2006, 140, 150–158. https://doi.org/10.1104/pp.105.070706

Into:

Li, J.; Quan, Y.; Wang, L.; Wang, S. Brassinosteroid Promotes Grape Berry Quality-Focus on Physicochemical Qualities and Their Coordination with Enzymatic and Molecular Processes: A Review. Int. J. Mol. Sci. 2022, 24, 445. https://doi.org/10.3390/ijms24010445

  1. Sasaki, Y.; Asamizu, E.; Shibata, D.; Nakamura, Y.; Kaneko, T.; Awai, K.; Amagai, M.; Kuwata, C.; Tsugane, T.; Masuda, T.; Shimada, H.; Takamiya, K.; Ohta, H.; Tabata, S. Monitoring of methyl jasmonate-responsive genes in Arabidopsis by cDNA macroarray: Self-activation of jasmonic acid biosynthesis and crosstalk with other phytohormone signaling pathways. DNA Res.2001, 8, 153–161. https://doi.org/10.1093/dnares/8.4.153

Into:

Sohn, S.I.; Pandian, S.; Rakkammal, K.; Largia, M.J.V.; Thamilarasan, S.K.; Balaji, S.; Zoclanclounon, Y.A.B.; Shilpha, J.; Ramesh, M. Jasmonates in plant growth and development and elicitation of secondary metabolites: An updated overview. Front. Plant Sci. 2022, 13, 942789. https://doi.org/10.3389/fpls.2022.942789

  1. Ban, T,; Shiozaki, S.; Ogata, T.; Horiuchi, S. Effects of abscisic acid and shading treatments on the level of anthocyanin and resveratrol in skin of kyoho grape berry. ActaHortic.2000, 514, 83-90. https://doi.org/10.17660/ActaHortic.2000.514.9

Into:

Shahab, M.; Silvestre, P.; Ahmed, S.; Colombo, R.; Roberto, S.; Koyama, R.; Souza, R. Relationship between anthocyanins and skin color of table grapes treated with abscisic acid at different stages of berry ripening. Sci. Hortic. 2020, 259, 108859. https://doi.org/10.1016/j.scienta.2019.108859

New reference was added:

  1. Shi, L.; Li, X.; Fu, Y.; Li, C. Environmental Stimuli and Phytohormones in Anthocyanin Biosynthesis: A Comprehensive Review. Int. J. Mol. Sci. 2023, 24, 16415. https://doi.org/10.3390/ijms242216415

Comments on the Quality of English Language

- The Language is weak, please correct for common grammar mistakes and incorrect sentence phrasing. Please see comments related to the “Abstract” section as an example for the entire manuscript.

Authors’ response: The remark has been accepted. The language was improved and checked.

Acknowledgments were added to the manuscript:

Acknowledgments: The manuscript presents the results carried out as part of the 'Double Hundred Talents Exchange' project in cooperation agreement between The National Institute of Horticultural Research, Skierniewice, Poland, and the Yantai Academy of Agricultural Science, Yantai China.
